# Lightweight Equivariant Graph Representation Learning for Protein Engineering

## Abstract

This work tackles the issue of directed evolution in computational protein design that makes accurate predictions of the function of a protein mutant. We design a lightweight zero-shot graph neural network model for multi-task protein representation learning from its 3D structure. Rather than reconstructing and optimizing the protein structure, the trained model recovers the amino acid types and key properties of the central residues from a given noisy three-dimensional local environment. On the prediction of higher-order mutations where multiple amino acid sites of the protein are mutated simultaneously, the proposed strategy achieves remarkably higher performance by 20% improvement at the cost of requiring less than 1% of computational resources that are required by popular transformer-based state-of-the-art deep learning models for protein design.

## 1 Introduction

Mutation is a biological process where the amino acid (AA) type of one or multiple sites of a specific protein is changed. While the wild-type proteins' functions do not always meet the demand of bio-engineering, it is vital to manually optimize the functionality, namely fitness, with favorable mutations so that they are applicable in designing antibodies (Wu et al., 2019; Pinheiro et al., 2021; Shan et al., 2022) or enzymes (Sato & Ishida, 2019; Wittmann et al., 2021).

A protein usually constitutes hundreds to thousands of AAs, where each residue belongs to one of twenty AA types. To optimize a protein's functional fitness, a greedy search is usually conducted in the local sequence, where AA sites are mutated to proper AA types to render a protein mutant with the highest gain-of-function (Rocklin et al., 2017). Such a process is called *directed evolution* Arnold (1998). To obtain a mutant with great fitness, multiple AA sites ($\sim$5-10) of the protein need to be mutated, namely *deep mutations* (see Figure 1). It, however, requires enormous experimental costs, as the total number of potential combinations of mutations for deep mutants is astronomical.

Since it is impossible to conduct systematic experimental tests on all possible deep mutations, *in silico* examination of protein variants' fitness becomes highly desirable. A handful of deep learning methods have been developed to accelerate the discovery of advantageous mutants. For instance, Lu et al. (2022) applied 3DCNN to identify a new polymerase with advantageous single-site mutation and enhanced the speed of degrading PET, i.e., a type of solid waste, by 7-8 times at 50°C. Luo et al. (2021) proposed ECNet that predicts functional fitness for protein engineering with evolutionary context. The model guides the engineering of TEM-1 $\beta$-lactamase and identifies variants with improved ampicillin resistance. Thean et al. (2022) enhanced SVD with deep learning to predict nuclease variants' activities in multi-site-saturated mutagenesis libraries from and identified Cas9 nuclease variants that possess higher editing activity of derived base editors in human cells.

Due to the scarcity of labeled protein data, researchers often pre-train an encoder for unsupervised learning with protein sequences or structures, and use the learned protein representations to train specific tasks, such as *de novo* protein design (Hsu et al., 2022), mutation effect prediction (Ingraham et al., 2019; Jing et al., 2020; Meier et al., 2021; Notin et al., 2022), and higher-level structure prediction (Elnaggar et al., 2021). In the context of fitness prediction of mutation effect, existing methods usually transform the problem to mini-*de novo* design, which infers a specific AA type from its microenvironment, or analogously its neighboring AA types. Current state-of-the-art sequence-based protein learning methods rely heavily on multiple sequence alignment (MSA; Riesselman et al. (2018); Frazer et al. (2021); Rao et al. (2021)) and protein language models (Elnaggar et al.,

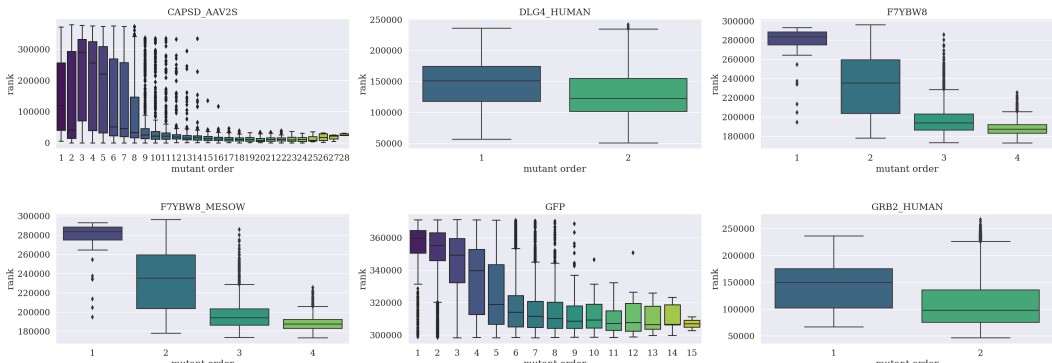

Figure 1: Mutating on more sites frequently results in a higher score, i.e., a smaller rank value.

2021; Rives et al., 2021; Nijkamp et al., 2022; Brandes et al., 2022). While MSA helps capture important evolutionary properties of the protein family, it nevertheless multiplies the requirements of computing resources. The latter protein language models derived from natural language processing (NLP) encode sequence semantics and often need hundreds of GPU cards to train on hundreds of millions of proteins. Meanwhile, an autoregressive inference process is usually required along the entire protein sequence to score a mutation on a single site, which further slows down the inference speed (Sato & Ishida, 2019; Liu et al., 2022; Hsu et al., 2022; Notin et al., 2022). More importantly, when predicting the fitness of the higher-order mutants, most of these models made a crude assumption that the mutations on different sites happen sequentially or individually, which is incorrect in most cases (Lehner, 2011; Breen et al., 2012). The ignored epistatic effects between different sites are potentially a key factor hindering the acquisition of favorable high-order mutants in directed evolution (Sarkisyan et al., 2016; Rollins et al., 2019).

Mutation of AA sites also occurs in nature, where an AA site might be mutated to any of the other 19 AA types in a random manner. It is suggested by natural selection that only the mutants that exhibit the best fitness and fit the environment survive. As a protein's functionality is determined by its structure, we encode the folded protein by a *protein graph* with AAs being graph nodes to provide an elegant 3D spatial description of the protein. The first-level information, such as AA types, spatial coordinates of $C\alpha$, and C-N angles between neighboring AAs, are embedded in node features. Altering AA types of a protein in nature can then be viewed as adding corruptions to the node features of the protein graph, and denoising the graph makes a remedy to search for mutants with the best fitness. We model the protein mutation effect prediction as a denoising problem with equivariant graph neural networks (Satorras et al., 2021). For a given protein, the recovered predictions can be leveraged to forecast the fitness of deep mutational effects and discover favorable mutants.

Compared to existing state-of-the-art deep learning methods for mutation effect prediction, such as ESM-1v (Meier et al., 2021) and ESM-IF1 (Hsu et al., 2022), the designed lightweight equivariant graph neural network (LGN) stands out in three perspectives.

First, **LGN improves generalization ability** through the multi-task learning strategy and biological prior knowledge. The pre-trained model encodes the chemical and physical properties of a given AA's microenvironment with domain knowledge for practically meaningful representations.

Secondly, **LGN avoids the independent-mutation assumptions** by generating the probabilities of all the amino acid residues at a time, which implements the joint distribution of all variations. In literature, the higher-order mutation effect is usually approached by summing up log-odd-ratio scores of the corresponding individual single-site mutants. The linear combination over separately assigned predictions is unsubstantiated, as the independent mutations neglect the epistatic effect.

Thirdly, **LGN is efficient in both the training and inference phases**. The spatial graph inputs portray the topological properties of proteins, which circumvents data augmentation that is typically required by sequence or grid representations. Equivariant message passing, alternatively, provides a feature distillation unit with translation and rotation equivariance and encodes AAs' microenvironment defined by the protein graph's geometry.

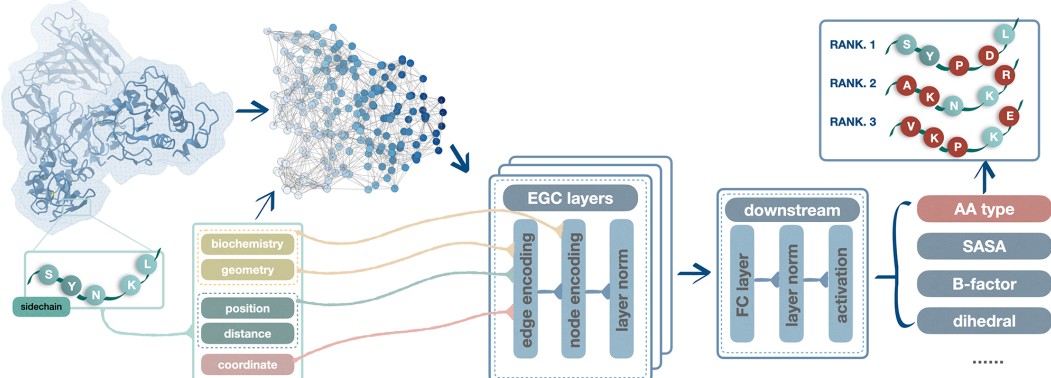

Figure 2: An illustration of the proposed LGN. The model is pre-trained with a set of protein graphs that are featured by (perturbed) node attributes and 3D positions with a multi-task learning strategy. A stack of EGC layers encodes rotation and translation equivariant structural representations for each node on individual graphs. Next, fully-connected layers are employed to learn different labels, where the AA type prediction is used for suggesting top-ranked mutations.

## 2   ZERO-SHOT LEARNING FOR PROTEIN RECOVERY

The excessive cost in laboratory results in scarce mutation scanning data, especially deep mutant results. It is thus favorable to pre-train a zero-shot protein prediction model that can be generalized directly to an unseen task without any further supervision to specialize the model.

### 2.1   GRAPH REPRESENTATION OF PROTEIN STRUCTURE

For a given protein, we create a k-nearest neighbor (kNN) graph $\mathcal{G} = (\mathcal{V}, \mathcal{E})$ to describe its 3D structure and molecular properties. Here each node $v_i \in \mathcal{V}$ represents an amino acid residue with $\boldsymbol{X} \in \mathbb{R}^{34}$ node attributes constituting biochemical properties and geometric properties of amino acids. The former includes 20-dimensional attributes of one-hot encoded amino acid types ($\boldsymbol{X}_{\mathrm{aa}}$), two scalars for each residue, i.e., solvent-accessible surface area (SASA) and the standardized crystallographic B-factor, and 5 normalized surface-aware node features. The geometric properties include the direction position ($\boldsymbol{X}_{\mathrm{pos}}$) of each residue by 3D coordinates of its $\alpha$-carbon and the relative position of the amino acid in the protein chain ($\boldsymbol{X}_{\mathrm{agl}}$) by the dihedral angles $\{\sin, \cos\} \circ \{\phi, \psi\}$ computed from the backbone atom positions. For a specific $v_i$ of the $i$th amino acid in the protein sequence, the dihedral angles are measured from $C\alpha_{i-1}, N_i, C\alpha_i, N_{i+1}$.

To build edge connections, we first define a symmetric adjacency matrix $\boldsymbol{A}$ with the kNN-graph to capture the nodes' microenvironment, i.e., each node is connected to up to $k$ other nodes in the graph that has the smallest Euclidean distance over other nodes, and the distance is smaller than a certain cutoff ($30\mathring{A}$). Consequently, if $v_i$ and $v_j$ are connected to each other, we have $\boldsymbol{A}_{ij} = \boldsymbol{A}_{ji} \neq 0$. The edge attributes $\boldsymbol{E} \in \mathbb{R}^{93}$ feature the connected edges in $\mathcal{E}$, including 15 inter-atomic distances, 12 local N-C positions, and the relative position in the protein sequence in 66-dimensions.

### 2.2   PRE-TRAINING WITH PRIOR DOMAIN KNOWLEDGE FOR BETTER PROTEIN FITNESS

The wild-type proteins suffer from random perturbations or mutations that not every AA site has the best AA type (Liu et al., 2022). To this end, we pre-train our model with a multitask learning strategy, which removes the natural corruptions and predicts key protein properties to help encode the microenvironment of the stabilized proteins of interest.

**AA Type Denoising**    We refine $\boldsymbol{x}_{\mathrm{aa}}$, the AA type a node, to $\tilde{\boldsymbol{x}}_{\mathrm{aa}}$ with a Bernoulli noise, i.e.,

$$\pi(\tilde{\boldsymbol{x}}_{\mathrm{aa}}|\boldsymbol{x}_{\mathrm{aa}}) = p\delta(\tilde{\boldsymbol{x}}_{\mathrm{aa}} - \boldsymbol{x}_{\mathrm{aa}}) + (1-p)\mathcal{M}(n, \pi_1, \pi_2, ..., \pi_n), \qquad (1)$$

where the confidence level $p$ is a tunable parameter that controls the proportion of residues that are 'noise-free'. The probability for the residue to become a particular type depends on the distribution

of the 20 types $\mathcal{M}(n, \pi_1, \pi_2, ..., \pi_n)$, which involves prior knowledge in molecular biology. This paper defines the distribution by the observed probability density of amino acid types in wild-type proteins [1]. See Appendix E to better understand the influence of different confidence levels.

**Geometric Properties Denoising**  For the continuous-valued features, such as 3D coordinates and dihedral angles, an i.i.d Gaussian noise is introduced, learning to remove which corresponds to approximating the data-generating force field of molecules (Zaidi et al., 2022). To be specific,

$$\tilde{\boldsymbol{x}}_{\text{pos}} = \boldsymbol{x}_{\text{pos}} + \sigma\epsilon, \quad \text{where } \epsilon \sim \mathcal{N}(0, I_3). \tag{2}$$

The noise effect is determined by $\sigma$, which is tunable to fit the scale of the noiseless raw feature.

**Bio-chemistry Properties Recovery**  Aside from denoising the perturbed residues type and geometric properties, other auxiliary tasks are introduced to help establish an expressive hidden microenvironment representation. Specifically, SASA is known to strongly influence AA type preferences, and B-factors are associated with the conformations and mobility of the neighboring AA. We thus introduce inductive biases to the model by predicting these two properties in the output.

**Label Smoothing with Amino Acid Substitution Matrices**  Protein sequence alignments provide important insights for understanding gene and protein functions. The similarity measurement of an alignment of protein sequence reflects the favors of all possible exchanges of one amino acid with another. We employ BLOSUM (Henikoff & Henikoff, 1992), a substitution matrix, to account for the relative substitution frequencies and chemical similarity of AAs. The matrix is derived from the statistics for every conserved region of protein families in **BLOCKS** database. As AA sites are more likely to be mutated to the AA type within the block of high similarity scores in the BLOSUM table, we hereby modify our loss function so that a mutation to an AA type with a higher similarity score accumulates a smaller penalty than to the one with a lower similarity score.

## 2.3 Protein Structure Representation with Equivariant GNNs

Proteins are structured in the 3-dimensional space, and it is vital for the model to predict the same binding complex no matter how the input proteins are positioned and oriented. Instead of practicing expensive data augmentation strategies, we follow Satorras et al. (2021) and construct SE(3)-equivariant neural layers for graph embedding. At the $l$th layer, an Equivariant Graph Convolution (EGC) inputs a set of $n$ hidden node properties embedding $\boldsymbol{H}^l = \{\boldsymbol{h}_1^l, \ldots, \boldsymbol{h}_n^l\}$ as well as the node coordinate embeddings $\boldsymbol{X}_{\text{pos}}^l = \{\boldsymbol{x}_1^l, \ldots, \boldsymbol{x}_n^l\}$ for a graph of $n$ nodes. The attributed edges are denoted as $\boldsymbol{E} = \{\ldots, \boldsymbol{e}_{ij}, \ldots\}$. The target of an EGC layer is to output a transformation on the node feature embedding $\boldsymbol{H}^{l+1}$ and coordinate embedding $\boldsymbol{X}_{\text{pos}}^{l+1}$. Concisely: $\boldsymbol{H}_{\text{pos}}^{l+1}, \boldsymbol{X}^{l+1} = \text{EGC}\left[\boldsymbol{H}^l, \boldsymbol{X}_{\text{pos}}^l, \boldsymbol{E}\right]$. To achieve this, an EGC layer defines

$$\begin{aligned}
\boldsymbol{m}_{ij} &= \phi_e\left(\mathbf{h}_i^l, \mathbf{h}_j^l, \left\|\mathbf{x}_i^l - \mathbf{x}_j^l\right\|^2, \boldsymbol{e}_{ij}\right) \\
\boldsymbol{x}_i^{l+1} &= \mathbf{x}_i^l + \frac{1}{n}\sum_{j\neq i}\left(\mathbf{x}_i^l - \mathbf{x}_j^l\right)\phi_x\left(\mathbf{m}_{ij}\right) \\
\boldsymbol{m}_i &= \sum_{j\neq i}\mathbf{m}_{ij} \\
\boldsymbol{h}_i^{l+1} &= \phi_h\left(\mathbf{h}_i^l, \boldsymbol{m}_i\right),
\end{aligned} \tag{3}$$

where $\phi_e, \phi_h$ are respectively the edge and node propagation operations, such as multi-layer perceptrons (MLPs). The $\phi_x$ is an additional operation that projects the vector embedding $\boldsymbol{m}_{ij}$ to a scalar value. The EGC layer preserves equivariance to rotations and translations on the set of 3D node coordinates $\boldsymbol{X}_{\text{pos}}$, while simultaneously performing invariance to permutations on the set of nodes $\mathcal{V}$ in the same fashion as GNNs.

---

[1]Retrieved from the folded protein dataset by **AlphaFold2** (Varadi et al., 2022) at `https://alphafold.ebi.ac.uk/`

## 2.4 Model Overview

Our model is depicted in Figure 2. We take a set of protein graphs with attributed nodes and edges, as well as each node's 3D coordinates, as the input to pre-train a zero-shot model. A stack of EGC layers is trained to extract rotation and translation equivariant representations for each node on individual graphs. The hidden representation is then sent to fully-connected layers to establish multiple outputs, such as AA type classification, SASA and B-factor prediction, and 3D coordinates denoising. The total loss for the multitask learning task is given by

$$\mathcal{L}_{\text{total}} = \mathcal{L}_{\text{aa}} + \lambda_1 \mathcal{L}_{\text{sasa}} + \lambda_2 \mathcal{L}_{\text{b-fac}} + \lambda_3 \mathcal{L}_{\text{pos}} + \lambda_4 \mathcal{L}_{\text{agl}}, \quad (4)$$

where $\lambda_i, i = 1, \ldots, 4$ are tunable hyper-parameters to balance different losses on auxiliary regression tasks. These losses are measured by mean squared error (MSE) loss. For AA type classification, we measure its loss $\mathcal{L}_{\text{aa}}$ by cross-entropy with label smoothing technique (Szegedy et al., 2016). The classification loss on an arbitrary node $i$ reads

$$\mathcal{L}_{\text{aa}} = (1-\varepsilon)\Big[-\sum_{y=1}^{20} p(y_{\text{aa}}|\boldsymbol{X}_i, \boldsymbol{E}_i) \log q_\theta(\hat{y}_{\text{aa}}|\boldsymbol{X}_i, \boldsymbol{E}_i)\Big] + \varepsilon\Big[-\sum_{y=1}^{20} u(y_{\text{aa}}|\boldsymbol{X}_i, \boldsymbol{E}_i) \log q_\theta(\hat{y}_{\text{aa}}|\boldsymbol{X}_i, \boldsymbol{E}_i)\Big],$$

where $p(y_{\text{aa}}|\boldsymbol{X}_i, \boldsymbol{E}_i)$ denotes the ground-truth distribution and $q_\theta(\hat{y}_{\text{aa}}|\boldsymbol{X}_i, \boldsymbol{E}_i)$ is the distribution of predicted labels following a softmax function. In order to improve the generalization and respect the prior biological knowledge, we modify the ground truth label distribution $p(y_{\text{aa}}|\boldsymbol{X}_i, \boldsymbol{E}_i)$ from the hard one-hot encoding to $(1 - \varepsilon)p(y_{\text{aa}}|\boldsymbol{X}_i, \boldsymbol{E}_i) + \varepsilon u(y_{\text{aa}}|\boldsymbol{X}_i, \boldsymbol{E}_i)$ when the predicted $\hat{y}_{\text{aa}} = y_{\text{aa}}$ and $\varepsilon u(y_{\text{aa}}|\boldsymbol{X}_i, \boldsymbol{E}_i)$ otherwise with some tolerance factor $\varepsilon$. In particular, we define the distribution of $u(y|x_i)$ by the BLOSUM substitution matrix.

## 3 Results

### 3.1 Experimental Setup

We train LGN on **CATH v4.3.0** (Orengo et al., 1997) with artificial noise to predict AA type, 3D coordinates, dihedral angles, and chemical properties (SASA and B-factor). The hidden embeddings of amino acids are learned by SE(3)-equivariant graph convolutions. The performance is validated by a zero-shot prediction task for the fitness of mutation prediction with deep mutational scanning (DMS; Fowler & Fields (2014)) datasets. The model performance is compared against popular state-of-the-art language models and structure-enhanced models.

**Baseline Models** We compare with a diverse of state-of-the-art models on the fitness of mutation effects prediction. In particular, DEEPSEQUENCE (Riesselman et al., 2018) trains VAE on protein-specific MSAs to capture higher-order interactions from the distribution of an AA sequence. MSA TRANSFORMER Rao et al. (2021) is a language model with aligned protein sequences of interest; ESM-1V (Meier et al., 2021) make zero-shot mutation predictions with masked language modeling; and ESM-IF1 (Hsu et al., 2022) predicts protein sequence with GVP (Jing et al., 2020), a graph representation learning methods for vector and scalar features of protein graphs. Furthermore, both TRANCEPTION (Notin et al., 2022) and PROGEN2 (Nijkamp et al., 2022) leverages autoregressive language models to retrieve AA sequence without family-specific MSAs.

**Lightweight Equivariant Graph Neural Networks (LGN)** To train our LGN framework, we first generate protein graphs for the sequences in **CATH**. See Appendix A for a detailed introduction to the generation, and Appendix C for a summary of the generated dataset. For the total number of $31,848$ protein graphs of $150$ nodes on average, we randomly pick $500$ graphs for validation and leave the remaining for model fitting. During the learning phase, we assign random perturbations to AA types and other features we mentioned earlier in Section 2. The noises are fixed to guarantee stable and comparable measurements at the validation step. The specific influence of different choices on the hyper-parameters (e.g., the noise level) will be discussed later in this section and Appendix D-E. The main architecture constitutes a stack of 6 EGC layers following 1 fully-connected layer to make predictions on the different learning tasks. On each node, the output is a vector representation consisting of 20 probabilities of the masked amino acid, 1 predicted SASA, 1 B-factors,

Table 1: Performance Comparison of Baseline Models on DMS assays prediction. Results with a higher Spearman's correlation are preferred.

| | DEEPSEQUENCE | TRANCEPTION | PROGEN2 | MSA TRANSFORMER | ESM-1V | ESM-IF1 | LGN (ours) |
|---|---|---|---|---|---|---|---|
| **CAPSD** | **0.4831** | 0.2307 | 0.2112 | **0.4419** | 0.2212 | 0.2170 | **0.3589** |
| **DLG4_HUMAN** | 0.5013 | **0.6200** | 0.5712 | 0.4654 | 0.4654 | **0.6164** | **0.6197** |
| **F7YBW8** | **0.3939** | **0.4280** | 0.3231 | 0.3483 | 0.2865 | 0.3714 | **0.4223** |
| **F7YBW8_MESOW** | **0.4205** | **0.4036** | 0.3231 | 0.3631 | 0.2522 | 0.3690 | **0.4210** |
| **GFP** | 0.6331 | **0.6647** | 0.6459 | **0.7069** | **0.7208** | 0.6203 | 0.6455 |
| **GRB2_HUMAN** | 0.3886 | 0.4441 | **0.5211** | 0.2808 | 0.3211 | **0.7033** | **0.6044** |
| average correlation | **0.4698** | 0.4177 | 0.4186 | 0.4511 | 0.3756 | **0.4714** | **0.5071** |

† The top three are highlighted by First, Second, **Third**.

and 4 dihedral values (when applicable). The 3D coordinates are derived directly from EGC outputs. The loss function by Equation 4 guides the backward propagation with ADAM (Kingma & Ba, 2015) optimizer. The model is trained with 300 epochs with the initial rate set to 0.001 and weight decay to 0.01. The learning rate is dampened to 0.0001 after 150 epochs.

**Evaluation**   All the models are evaluated with deep mutational scanning (DMS) assays that assess a diverse set of 15 proteins, where 9 of them only contains single-site mutation scores, and 6 of them have both single-site and higher-order mutational records (see Appendix B). The protein structures are folded from the provided sequence information with ALPHAFOLD2 (Jumper et al., 2021), following the exact same pre-processing steps as in **CATH** for generating protein graphs. The only difference is that we do not append artificial noises onto the test proteins, as we assume they are already noisy. We then send the unmutated test proteins to the pre-trained LGN model and use the log-odd-ratio in Equation 5 of the predicted probabilities of AA types for suggesting the rank of deep mutations. The prediction performance is evaluated on Spearman's correlation coefficient between the computational and experimental scores on all the mutation combinations.

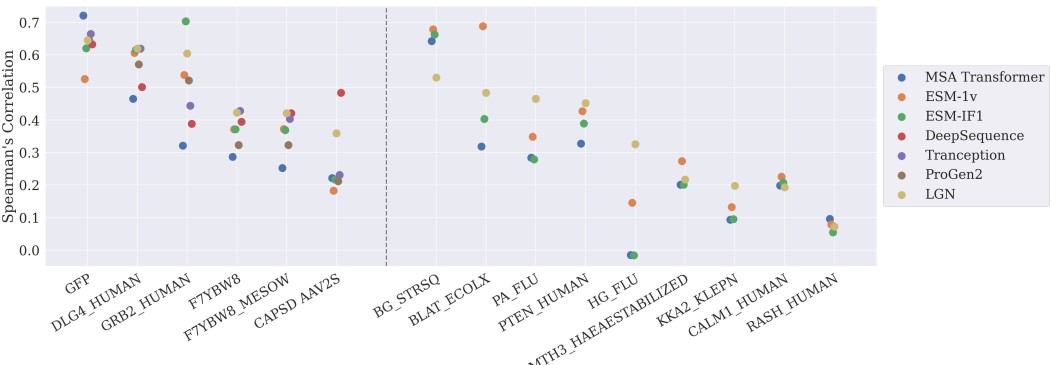

Figure 3: Per task performance on the fitness of deep mutant prediction with pre-trained models. Each point indicates Spearman's correlation coefficients on the corresponding protein. The left 6 proteins contain higher-order mutations, and the right 9 proteins record shallow mutants.

## 3.2   FITNESS OF DEEP MUTANTS PREDICTION

The first experiment evaluates the fitness of proteins' mutation effects prediction, where the fitness scores are inferred directly from a pre-trained model without supervision on a task-specific model. We visualize the overall performance comparison on protein-wise Spearman's correlation coefficients in Figure 3. The deep mutation scores are reported in Table 1, where LGN outperforms baseline methods and achieves at least comparable results in single-site mutant tests. Overall, our model achieves 0.5071 weighted average correlations on deep mutant effect predictions. While DEEPSEQUENCE achieves superior performance over the majority rest, it should be noticed that the model has to be trained on every new protein, and it cannot be generated for other proteins. Also, the training speed and performance of the learned model rely heavily on the quality of the available MSA information, which can very a lot on different proteins.

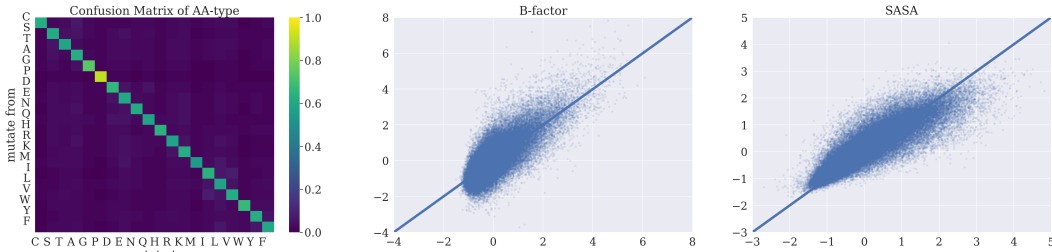

Figure 4: The three plots (from left to right) display the confusion matrix of predicted AA types, and linear regression on the predicted SASA and B-factor, respectively.

### 3.3 PROTEIN RECOVERY

This experiment investigates the auxiliary learning tasks of the pre-trained model, including the learning performance in predicting AA types, SASA, and B-factor. In specific, we visualize the confusion matrix of the predicted AA types with respect to the ground-truth AA types to see evaluate the model's capability to recover from noisy sequences to the original sequence, i.e., if the large values are accumulated to the diagonal of the confusion matrix. For the SASA and B-factor predictions, we examine the $R^2$ of the predicted and the ground-truth values on **CATH**. The results are visualized in Figure 4 with denoised AA type, as well as the predicted SASA and B-factor as the output tasks. The AA type prediction achieves high accuracy with the majority of predictions accumulated on the diagonal line. For the two regression tasks, we fit the true value and the predicted value with linear regression. The estimated coefficients are 1.008 and 0.989 for B-factor and SASA, respectively. The $p$-value for both coefficients is $< 0.001$. In addition, Pearson's correlation coefficients for the two variants are respectively 0.884 and 0.791.

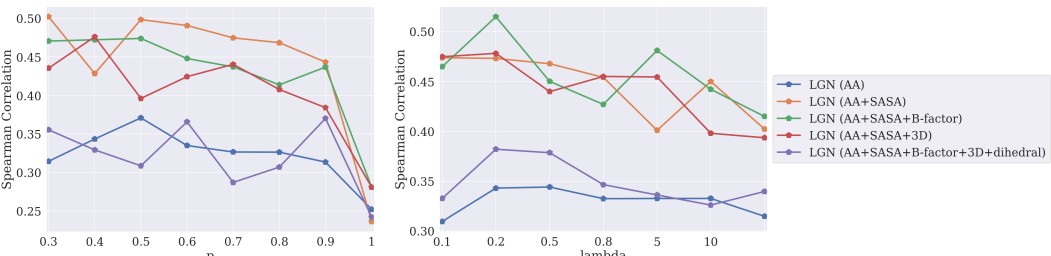

Figure 5: Average Performance with different $p$s (left) and $\lambda$s (right) on the auxiliary tasks.

### 3.4 SELECTION CRITERIA ON THE ADDITIONAL HYPER-PARAMETERS

As LGN introduces two additional hyper-parameters in training the model, this section examines the influence of selecting different $p$s and $\lambda$s. Figure 5 visualizes a selection of model performance on the Spearman's correlation with variant $p$s and $\lambda$s with wild-type noise distribution.

While the choice of $p$ can be determined by prior knowledge regarding the quality of wild-type proteins, here we treat $p$ as a data-driven hyper-parameter to be optimized during the model training. We exclude extremely small $p$s to avoid drastic perturbation rates and search for the optimal $p \in \{0.3, 0.4, \ldots, 0.9, 1\}$. The different choices on $p$s are validated with different learning tasks on the left side of Figure 5 for higher-order mutants. In general, a moderate $p$ between 0.3 and 0.6 best suits the majority selection of learning modules and noise distribution. Based on the overall performance, we suggest $p = 0.6$ as the default value of the confidence level (See Appendix E for more results on different types of perturbation noise and proteins).

We also investigate a wide range of the choices of $\lambda$s. For simplicity, we let $\lambda_1 = \lambda_2 = \lambda_3 \in \{0.05, 0.1, 0.2, 0.5, 0.8, 5, 10\}$ and fix $\lambda_4 = 0.5$. All the results are conducted under the recommended $p = 0.6$ with wild-type noise. We report the average performance on deep mutants in the

Table 2: Comparison of baseline models. The train and inference speed is tested on **GFP**.

| model | DEEPSEQUENCE | MSA TRANS. | ESM-1V | ESM-IF1 | TRANCEPTION | PROGEN2 | LGN (ours) |
|---|---|---|---|---|---|---|---|
| input | sequence | sequence | sequence | sequence+structure | sequence | sequence | structure |
| MSA | ✓ | ✓ | | | ✓ | | |
| train on new protein | ✓ | | | | | | |
| training dataset | - | **Uniref50** | **Uniref90** | **CATH+AF2** | **Uniref100** | **Uniref90+BFD30** | **CATH** |
| | - | (2018-03) | | (2020-03) | | | v4.3.0 |
| training size (M) | - | 45 | 98 | 12 | 249 | $> 1,000$ | 0.03 |
| max. input token | | 1,024 | 1,024 | 1,024 | 1,280 | 1,024 | $2,687^1$ |
| # parameters (M) | 4.3 | 100 | 650 | 142 | 700 | 2,700 | 1.5 |
| # layers | 1,600 | 12 | - | 20 | 36 | 32 | 6 |
| # head | - | 12 | - | 8 | 20 | 32 | - |
| # hid. dim. | $100 - 2,000$ | - | - | $512 - 2,048$ | - | - | 512 |
| speed (training day) | - | $13^2$ | 6 | 653 | $\sim 100$ | - | 0.17 |
| resource (train) | - | $128 \times$V$100^2$ | $64 \times$V100 | $32 \times$V100 | $64 \times$A100 | $? \times$TPU-v3 | $1 \times$3090 |
| preparing speed (sec) | $6,360 + 25,020$ | 6,360 | - | - | 6,360 | - | - |
| inference speed (sec) | 608 | 927 | 75 | 102 | 1,920 | 1,440 | 25 |

right plot of Figure 5, which demonstrates a relatively flat and steady trend with a mild peak at $\lambda = 0.2, 0.5$. Additional results are provided in Table 8 of Appendix E with various model setups.

## 3.5 INFERENCE SPEED

LGN consumes significantly fewer computational resources in training and inference. We compare the model scale, inference time, and prediction performance in Figure 6 and Table 2.

The model size and the required resources with the baseline methods are provided by the authors. As the majority of models are pre-trained, we record the inference speed on a single 3090 GPU. While the time cost is significantly lower than experimental methods, we measure it to indicate the cost of forward propagation in one iteration, which can be viewed as the indirect empirical evidence of the training cost. As each protein requires independent inference progress, we hereby take **GFP** as an example protein sample. The protein constitutes 236 amino acid residues, and it has over 50,000 mutant records (see Table 3 in Appendix B for more details). Note that: 1). The 2,687 input token length only refers to the maximum protein length we used during training. In fact, the model itself can process large protein graphs containing over tens of thousands of amino acids. 2). The training speed and required resources for MSA TRANSFORMER are retrieved from Meier et al. (2021). The original work by Rao et al. (2021) only reports that they used 32×V100 GPUs for training, without revealing the training time.

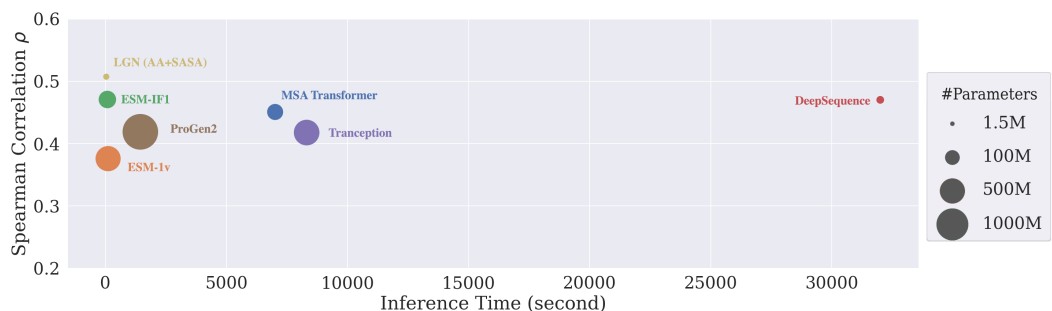

Figure 6: Comparison of Inference Efficiency. The area of the ball indicates the number of network parameters of a model. Our model (in blue) can achieve SOTA performance (y-axis) with minimum inference time (x-axis) and 1% number of parameters of the ESM.

## 4 RELATED WORK

**Protein Sequence and Structure Representation** Due to the enormous experimental cost of measuring protein structures, the number of known protein sequences is thousands of times larger than

protein structures (Eswar et al., 2008; Hsu et al., 2022). Meanwhile, the protein sequences representation is highly similar to human language, which naturally promotes the fast development of natural language processing (NLP), especially transformer-based methods for encoding protein sequences (Coin et al., 2003; Meier et al., 2021; Ofer et al., 2021; Castro et al., 2022). However, the geometry of proteins also suggests higher-level structures and topological relationships that are vital to protein functionality. Structure prediction of proteins always attracts great attention in the field (Chi & Liberles, 2016; Jumper et al., 2021; Baek et al., 2021; Varadi et al., 2022). The breakthrough progress in protein folding also enriches structured proteins. For instance, Hsu et al. (2022) and Ma et al. (2022) mixed experimentally-tested and ALPHAFOLD-predicted for model training, which greatly eases the data shortage problem and achieves significant performance gain.

**Structural encoding for Protein Graphs** According to the laws of physics, the atomic dynamics do not change no matter how a protein is translated or rotated from one place to another (Han et al., 2022). Therefore, the inductive bias of symmetry should be incorporated into the design of protein structure-based models. To this end, research work has been proposed to respect the spatial relationship of amino acids (Torng & Altman, 2017; Sato & Ishida, 2019). Such CNN-based methods aggregate the local structure of each residue and integrate estimated local qualities into the whole protein properties. However, these methods neglect geometric equivariance, which can usually be captured by equivariant graph neural networks (Ganea et al., 2021; Stärk et al., 2022).

**Protein Representation** As existing protein language models require high computational costs and are difficult to train, finding an effective feature representation of protein data is important for downstream tasks (Thompson et al., 2012). Contrastive learning and self-prediction (Elnaggar et al., 2021; Zhang et al., 2022; Hsu et al., 2022) used self-supervised pre-training methods to extract good representation for reducing computational resources. Despite only applying classical representation learning methods on protein, some researchers designed sophisticated encoders for expressive protein representation. For instance, Li et al. (2022) proposed $\vec{W}$-GNN variants that efficiently interact with scalar-vector features. Somnath et al. (2021) introduced HOLOPROT to connect different modalities of proteins, including surface, structure, and sequence representation.

**Mutation Effect Prediction** Multiple sequence alignment (MSA) is an essential ingredient for many of the existing state-of-the-art methods to predict the effect of single amino acid substitutions such as DEEPSEQUENCE (Riesselman et al., 2018), ALPHAFOLD2 (Jumper et al., 2021), MSA TRANSFORMER (Rao et al., 2021), and LM-GVP (Wang et al., 2022). The MSA for a protein sequence or domain captures meaningful information on the evolutionary information of the protein within its family at the cost of bringing severe limitations–not all proteins are alignable, such as CDRs of antibody variable domains (Shin et al., 2021), and not all the alignments are deep enough to train models sufficiently large to learn the complex interactions between residues. To deal with this issue, ESM-1V (Meier et al., 2021) trains a zero-shot model on a large set of unaligned sequences to secure a scalable and bias-free training procedure, and TRANCEPTION (Notin et al., 2022) leverages autoregressive predictions and retrieval of homologous sequences at inference.

## 5 CONCLUSION

Designing directed evolution on proteins, especially with deep mutants for functional fitness, is of enormous engineering and pharmaceutical importance. However, existing experimental methods are economically costly, and *in silico* methods require significant computational resources. This paper proposed a lightweight zero-shot model for mutant effect prediction on arbitrary numbers of AAs by transferring the problem to denoising a protein graph. Our model is trained to recover AA types and other important properties (e.g., B-factor, SASA, and the spatial position of C$\alpha$) from observed noisy proteins. We employ translation and rotation equivariant neural message passing layers to extract geometric-aware representation for the microenvironment of central AAs and thus grasp rich information for efficiently learning protein function. The model achieves state-of-the-art performance on PDB datasets in deep mutant tests with significantly fewer computational resources than existing SOTA models.

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
