# OpenReview forum: "Lightweight Equivariant Graph Representation Learning for Protein Engineering"
_ICLR.cc/2023/Conference — Submitted to ICLR 2023_

### Official Review · Reviewer_G8po · 2022-10-24

**Confidence:** 5
**Correctness:** 2
**Technical Novelty And Significance:** 2
**Empirical Novelty And Significance:** 2
**Recommendation:** 3

**Clarity, Quality, Novelty And Reproducibility:**

The authors introduced a new approach to the protein design problem but did not present sufficient evidence to support the claimed advantages. The results also lack further explanation and interpretation in detail. As a result, it is unclear to judge whether the contributions are valid. In addition, no clear explanation of how the model could be used is provided.

**Strength And Weaknesses:**

# Strength
1. The authors provided a clear formulation of the problem of interest.
2. The authors provided a clear statement of their claimed major contributions.
3. The authors provided a very detailed description in the method section, including intuition and mathematical formulation of the loss function.
# Comments
1. The authors did not offer a very strong intuition on applying zero-shot or multi-task learning to the mutation fitness prediction problem.
2. The authors claimed that the model improved the generalization ability, but no clear evidence is given to support how the generalization ability has been improved. There are also no explicit explanations for this supposedly most significant advantage of the model.
3. There is not any reference or explanation to Figure 2 in the main text, which puts the reader in a quite confusing situation of wondering about its purpose.
4. The authors tried various LGN configurations and presented the results in Figure 3. However, there isn’t a conclusion or hypothesis on which configuration may result in better performance under certain conditions. The authors could have conducted more experiments and provided more insights into these results.
5. Figure 3 is not very illustrative, e.g., ESM-1v is missing/hidden for F7YBW8, and the labels for the LGN models are not distinguishable.
6. The comparison in section 3.5 is somehow not clear enough. First, the authors only tested on one protein sample, which does not offer sufficient evidence to prove the claimed advantage. Second, the training time usage comparison is ambiguous since different models are trained on datasets of various sizes, as indicated in Table 1 of the paper.
7. The authors did not mention how the model might be used for downstream tasks, particularly mutation prediction. I would personally be interested to know whether there would be a web tool or a pre-trained model for users to try.
# Minor
1. Some grammar mistakes (although without affecting the understanding of the text), such as “approach … sum up” and “results are conducted”.
2. Since Table 2 is only included in the supplementary material, the “Table 2” reference in the main text’s section 3.5 might need be changed.


**Summary Of The Paper:**

The authors studied the problem of amino-acid mutation prediction in protein by regarding this problem as solving a denoising problem with lightweight GNN. Several large-scale pre-trained models have previously been proposed and applied to this problem, such as ESM-1v and MSA transformer. The authors proposed a new lightweight graph neural network approach for this problem. Three major contributions are claimed.
1. Improvement in generalization ability.
2. Improvement in efficiency.
3. Improvement in performance on high-order mutations.

**Summary Of The Review:**

The authors introduced a new approach to the protein design problem but did not present sufficient evidence to support the claimed advantages. The results also lack further explanation and interpretation in detail. As a result, it is unclear to judge whether the contributions are valid. In addition, no clear explanation of how the model could be used is provided. In my opinion, the paper does not meet the standard of ICLR at this moment and may need further major revisions with a focus on the results part.

---

> ### Author Response · Authors · 2022-11-05
> **Response to Reviewer G8po (2)**
>
>
> 6. comment6 - **Inference Speed Comparison**:
>
> We respectfully disagree with the reviewer. Regarding "testing on one protein does not provide enough evidence", the computational cost of all baseline models is massive even by solely reading Table 1 and by reviewing those models' architectures (MSA extractor and attention models are generally required, and both of them are expensive beasts). Secondly, the inference time is not directly aggregatable among different proteins, as they have different AA sequence lengths, not to mention that the number of MSA sequences varies a lot on different source proteins. As such, taking an average over the inference time of different proteins does not make much sense.
>
> Also, we would like to clarify that Figure 6 reports **INFERENCE** time on an unseen protein, instead of **TRAINING** time on the training dataset. This new protein is fixed to all the baseline models where the raw input is usually provided in a text file of *.pdb*, *.a2m*, or *.a3m* format. As the inference time has nothing to do with the size of the training dataset, we do not see any unfairness in comparing the inference time of our model with other baselines (on the same protein). Furthermore, if our model can achieve comparable or higher performance with a smaller training dataset, it substantially proves our model is more expressive and more suitable to the research task of interest than the baseline methods.
>
>
> 7. comment7 - **downstream task and model playground**:
>
> The full model architecture for a specific (e.g., fitness prediction of deep mutations) is visualized in Figure 1, fully described in Section 3.4, and supplemented in Appendix B.2 regarding the predicted fitness score. In summary, for a new unseen protein, we transform it into a protein graph and send it to the pre-trained model to predict the target labels, including the probabilities of AA types. The $20$-dimensional probabilities then calculate the log-odd-ratio by eq.5 in Appendix B.2 as the predicted fitness score, which is a standard transform step in deep mutation models.
>
> We thank the reviewer for their interest in our method and appreciate the suggestion for releasing it as a web tool for the community. We will make our efforts to realize it.
>
>
>
> 8. minor1 - **grammartical mistakes**:
>
> Thank you for identifying the mistakes. We have gone through the entire text carefully and fixed the identified typos.
>
>
> 9. minor2 - **Table2**:
>
> We have modified the expression accordingly to direct the Table and avoid ambiguity. It now reads: " see Table 2 in Appendix B for more details".
>
>
> 10. clarity and quality:
>
> We thank the reviewer for affirming the clearness of our problem formulation, the methodology explanation, and the intuition of the proposed method. We regret that the reviewer does not agree with the interpretation of the experimental results and the explanation of the model details. We have responded to the posted concerns by the reviewer point by point and hopefully, they can answer the questions or enhance the readability of the paper. We also submitted a revised version following the kind suggestions of all the reviewers. We appreciate it if the reviewer could revisit the updates and we are more than happy to continue the discussion.

---

> ### Author Response · Authors · 2022-11-14
> **Response to Reviewer G8po**
>
> We thank the reviewer for their feedback. Below we respond to the concerns point by point.
>
> 1. comment1 - **intuition for multi-task and zero-shot learning**:
>
> Employing zero-shot prediction for protein learning tasks was first formally discussed by Rao et al. (2021), which "transfer of a model to a new task without any further supervision to specialize the model to the task". In other words, “if deep protein models can learn the information necessary to solve a task from pre-training, then they can be applied directly to new instances of the task, without specialization. This would mean that in practice a single general-purpose model can be trained once and then applied to a variety of possible tasks.” Since mutation scanning data, especially deep mutant data, is considerably scarce, training a zero-shot model has been practiced by more and more research works. We have added a brief explanation at the beginning of Section 2, which reads "The excessive cost in laboratory results in scarce mutation scanning data, especially deep mutant results. It is thus favorable to pre-train a zero-shot protein prediction model that can be generalized directly to an unseen task without any further supervision to specialize the model. ".
>
> Regarding the multi-task learning strategy, as the zero-shot prediction does not specify the downstream task, it is favorable to establish additional learning targets to enhance the expressivity of the encoded representation, so that they are capable of unseen new tasks. In the main text, we have justified the design by "a multitask learning strategy...removes the natural corruptions and predicts key protein properties to help encode the microenvironment of the stabilized proteins of interest." in Section 2.2.
>
>
>
> 2. comment2 - **generalization ability of the pre-trained model**:
>
> The fundamental idea of zero-shot prediction is to train a general model that nails unseen tasks without further supervision. Specifically, LGN is pre-trained on multiple tasks to recover the key properties of the protein, such as the AA sequence, SASA, and b-factor. Later, the trained model is applied directly to DMS prediction, which is never seen before. We show that our model achieves satisfying performance on a variety of proteins, and the prediction scores support the generalization ability with the most direct and intuitive evidence.
>
>
> 3. comment3 - **figure 2**:
>
> We displayed Figure 2 to support the fact that "mutating on more sites would result in a higher score.", and we stated this in the caption of the Figure. We have added a reference on page 1 when introducing deep mutants: "Multiple AA sites (~5-10) of the protein need to be mutated to obtain a mutant with great fitness (see Figure 1)..." and bring the figure forward at the top of page 2.
>
>
> 4. comment4 - **optional learning objectives**:
>
> In this paper, we pre-trained our graph on the 31,848 proteins from CATH v4.3.0 that have less than 40% sequence identity and the test proteins are folded with AlphaFold 2. Under this construction, our empirical results show that interpreting the AA type, SASA and B-factor is a solid combination. Meanwhile, we define the objective function (4) to show other available prediction targets, such as the dihedral angle, which might result in better performance with a different training dataset. Due to the length limit, we left additional experimental results that compare different choices of learning objectives in Appendices.
>
>
> 5. comment5 - **clarity of Figure 3**:
>
> We have updated the Figure to display scores from one LGN variant to compare against baseline methods. In addition, a direct numerical comparison of the models' average performance of different proteins for DMS prediction is reported in Table 1 on page 6.

---

### Official Review · Reviewer_LEve · 2022-10-24

**Confidence:** 4
**Correctness:** 1
**Technical Novelty And Significance:** 2
**Empirical Novelty And Significance:** 1
**Recommendation:** 5

**Clarity, Quality, Novelty And Reproducibility:**

There is a lack of clarity in the writing.  The results and presentation could be highly improved in quality.

The premise of the work is novel but not sure that it will yield good results in practice. Most comparisons are to work that does not use structure information.

**Strength And Weaknesses:**

Strengths:

The general aim of leveraging structure information to improve fitness prediction performance in the context of protein design is important.

Weaknesses:

I am not sure if the overall premise of treating mutants as perturbations to node signals and using a denoising autoencoding approach works for their goals. The premise of denoising, is that one brings the perturbed data back to the original data manifold, so likely perturbations will simply be denoised to the training distribution (consisting primarily of wild type proteins). This will not allow for novel protein generation. Indeed there is no quantification in the manuscript for the ability to generate fitter more novel mutants.  The authors state “Altering AA types of a protein in nature can be viewed as adding corruptions to the node features of the protein graph, and denoising the graph makes a remedy to search for mutants with the best fitness”  The connection between denoising and fitness is assumed but never fleshed out.


Additional comments for improvement are below:

1. Sato & Ishida (2019) are cited twice in the introduction. I believe the first citation in the following sentence is incorrect: “Sato & Ishida (2019) applied 3DCNN to identify a new polymerase with advantageous single-site mutation and enhanced the speed of degrading PET by 7-8 times at 50”. The correct citation should be Lu et al., Nature, vol. 604, pgs 662–667, 2022. Note that the temperature of 50 degrees is measured in Celsius, and some background information (i.e. PEG is a waste product that can recycled by degradation) would be helpful to contextualize this work. The second citation of this paper, regarding autoregressive inference for scoring single-site mutations, is also irrelevant. Please re-check all citations, taking care to only include works that are relevant to this paper.

2. What does “capture the micro-frame property in the protein graph geometry” mean? Does it refer to local geometric properties (e.g., alpha helix, beta sheets) of a contiguous subset of residues in the protein? The term ‘micro-frame’ is never clearly defined.

3. The usage of terms ‘invariance’ and ‘equivariance’ is inconsistent throughout.

4. The 93-dimensional edge attributes include a 66-dimensional encoding of the ‘relative position in the protein sequence’. This comprises of a 65-dimensional 1-hot encoding of sequence distance between pairs of nodes, and a scalar contact signal describing ‘if the two residues are in contact in space’. Here, the threshold value of 65 (possible typo in Appendix A.3 states 64) is chosen according to the sequence distance distribution. Is this threshold fixed for all proteins in your dataset?

5. In the introduction, the authors state that their model predicts mutation scores without assuming independence of individual mutations. Specifically, the authors point to the shortcomings of summing up log-odd ratios. However, in Appendix B.2, Eqn 5, the fitness score for multi-site mutations is computed by summing log-odd ratios for single- site mutations. Can you please clarify?

6. “The p-value for both coefficients is 0.000.” This is likely due to rounding or truncation. Perhaps say that p-values for both coefficients is < 0.001?

7. I cannot find any discussion on the limitations of this work. The proposed model is ‘lightweight’ and efficient to train because it relies on the availability of prior knowledge, i.e., tertiary protein structures. This is an important distinction as the models argued against in the introduction are large due to their attempt to model evolutionary-scale information about protein sequence composition in the absence of structure information.

8. Can you comment on the application of this model towards predicting protein stability changes (i.e., changes in the Gibbs free energy of unfolding) between wild type and mutant proteins?

9. Consider citing related work on fitness prediction using regularized latent space optimization (Castro et al., Nature Machine Intelligence, vol. 4, pgs 840–851, 2022), and structure-informed sequence embedding (Wang et al., Scientific Reports, vol. 12, 6832, 2022).

10. “The latter protein language models derived from natural language processing (NLP) encode sequence semantics and often need hundreds of GPU cards to train on billions of protein sequences” This claim needs a citation as most protein datasets are upper bounded at 100s of Millions. ESM-1 (Rives et al 2021) was trained on 250M sequences and AlphaFold 2 (Jumper et al 2021) trained on 350M.

11. Figure 2 seems to be an important result however it is never mentioned in the manuscript and therefore is difficult to interpret.

12. The ablations in Figure 3 are obscured by their same coloring, especially for datasets where their y-positions overlap.

13. I would also suggest adding quantifications of the ability to perform directed protein evolution (i.e. generate newer fitter molecules).



After reading the rebuttal:
I raised the score from "reject, not good enough" to "marginally below the acceptance threshold":
They have made all the corrections that we and the other reviewers pointed out.
They have included additional comparisons to DeepSequence, Tranception, ProGen2 in Figure 3, Figure 6, and Table 1. In cases where a comparison could not be performed (EVE and Potts model) due to technical issues, the authors provide a reasonable explanation.
The wording around "independence of single mutations" (Appendix B.2) that was confusing for us and the other reviewers has been fixed.
I hesitate to recommend acceptance because:
VAE type models can generate high fitness protein sequences, and they do not suffer from the independence of single mutations assumption. Our main critique, i.e. "This will not allow for novel protein generation. Indeed there is no quantification in the manuscript for the ability to generate fitter more novel mutants." was not addressed in the rebuttal. Ultimately their method is good for predicting fitness and other properties efficiently, but is not practically useful for generating new sequences.
The authors admit that computing MSA for pre-training is computationally expensive. However they assume that this info is given ahead of time and they do not include it in their runtime. At the same time they argue: "we doubt the speed of inference time for Potts Model, as the efforts in preparing the MSA information should also be included."


**Summary Of The Paper:**

The authors propose a ‘lightweight graph neural network’ (LGN) for fast zero-shot fitness prediction of mutational effect from protein structures. Noisy perturbations of node and edge features computed from the protein structure are fed into Equivariant Graph Convolution (EGC) layers to extract hidden representations. These representations are used for multitask learning of amino acid type, microenvironment properties (solvent accessible surface area and crystallographic B-factor) and denoising of residue coordinates. Using the learned representations, the authors demonstrate highly efficient SOTA performance in mutation effect prediction.

**Summary Of The Review:**

The general aim of leveraging structure information to improve fitness prediction performance in the context of protein design is important. However the manuscript and its approach are not well motivated or clearly presented. Moreover, the manuscript contains errors that impact both the strength and the readability of the work presented. The captions of Figures are sparse and make it difficult to interpret results and one figure is not mentioned anywhere in the main manuscript.

---

> ### Author Response · Authors · 2022-11-14
> **Response to Reviewer LEve (2)**
>
> 8. weakness8 - **predicting changes of Gibbs Free Energy of Unfolding**:
>
> It is possible for the model to apply to particular tasks, such as predicting $\Delta\Delta G$ for stability mutational designs, which is one of our future plans to attempt. The most straightforward approach is to send the hidden representation of the wild-type protein and mutant protein from the second last layer of the LGN layer to other trainable layers (such as FC-layers) to learn a predicted $\Delta G$, from which we use a non-linear transform (a final FC-layer, for instance) to predict the $\Delta\Delta G$. Another possible route is to use the labeled dataset (i.e., proteins and protein mutations with given $\Delta G$) to fine-tune the pre-trained model, where the prediction target would accordingly be changed to $\Delta G$.
>
> While we discuss some preliminary ideas for solving this specific task, we are aware that further considerations on the model design might be required in order to make accurate predictions on the $\Delta\Delta G$, as the labeled data is relatively scarce, and the unavoidable differences on the measured Gibbs free energy from different sources (e.g., tested by different labs) might address additional difficulties in this learning task.
>
>
> 9. weakness9 - **additional citations**:
>
> Thank you for supplementing additional references. We have added the two works in the *Related Work* section accordingly. ReLSO (Castro et al., 2022) was added to the first paragraph of transformer-based methods for encoding protein sequences; LM-GVP (Wang et al., 2022) is cited in the last paragraph of mutation effect prediction that "Multiple sequence alignment (MSA) is an essential ingredient ..., such as..., and LM-GVP."
>
>
> 10. weakness10 - **size of protein datasets**:
>
> Thank you for identifying this problematic expression. While extra-large datasets have been considered by some studies (for instance, ProGen2 used 2.7B and 6B proteins to train the two largest models in the paper), we agree with the reviewer that hundreds of millions of proteins are more frequently used in literature. We have updated the statement to "...train on hundreds of millions of protein sequences." to avoid ambiguity.
>
>
>
> 11. weakness11 - **Figure 2**:
>
> We displayed the figure to support the fact that "mutating on more sites would result in a higher score.", and we stated this in the caption of the figure. We have added a reference on page 1 when introducing deep mutants: "Multiple AA sites (~5-10) of the protein need to be mutated to obtain a mutant with great fitness (see Figure 1)..." and bring the figure forward at the top of page 2.
>
>
> 12. weakness12 - **colors in Figure 3**:
>
> Thank you for identifying the readability issue. We have adjusted the Figure to display only one of our variants to compare against baseline methods. We also change the x-axis slightly to avoid over-crowded markers. In addition, a direct numerical comparison of the models' average performance of different proteins is added in the last row of Table 1 on Page 6.
>
>
> 13. weakness13 - **other quantifications**:
>
> In the paper, we investigated the fitness of auxiliary tasks to show the learning performance of the pre-trained model from different perspectives (aside from the specific task, i.e., DMS prediction). For a particular task, another measurement of interest might be the success rate of the mutation in wet lab tests, and it is a critical indicator for in silico designs. However, to the best of our knowledge, existing measurements are generally qualitative (for instance, investigating the physical-chemical properties of the mutants) and they usually serve as a filter. We appreciate the reviewer for bringing up this new perspective and we will keep an eye on it.
>
>
> 14. quality and novelty:
>
> We thank the reviewer for affirming the novelty of our work. We have made efforts in polishing our work, and the revised version is now updated on this page. We appreciate it if the reviewer could check our response as well as the latest version of the submission, and we are happy to answer any further concerns addressed by the reviewer.
>
> Regarding the second doubt, we argue that structure-based methods for directed evolution are still at their earlier stage of development. It is undeniable that language models have been researched more throughout than geometric deep learning approaches, and protein sequences are more abundant and reliable than structures. Consequently, existing sequence-based models have a higher volume than structure-based models (especially in the context of zero-shot predictions). However, it is well-known that protein function is determined by its structure, which is further determined by its AA sequence.
> Ultimately, directed evolution studies the functional revision of proteins, including their geometry, is arguably an essential choice for *in silico* methods.

---

> ### Author Response · Authors · 2022-11-14
> **Response to Reviewer LEve (1)**
>
> We very much appreciate the detailed reviews and constructive feedback! We answer each of the concerns in detail.
>
>
> 1. weakness1 - **citation issues in Introduction**:
>
> We follow the reviewer's suggestions to fix the citation issues. For 3DCNN, we have changed the presentation in the *Introduction* section accordingly, which now reads: "Lu et al. (2022) applied 3DCNN to identify a new polymerase with advantageous single-site mutation and enhanced the speed of degrading PET, i.e., a type of solid waste, by 7-8 times at 50$^{\circ}\text{C}$.". Regarding the second citation, we have deleted ESM-1v under the category of mutation scoring methods with autoregressive inference.
>
>
> 2. weakness2 - **micro-frame property of a protein graph**:
>
> When constructing the protein graph, we extract the geometric information of the underlying protein. As detailed in Appendix A.2, each node (amino acid) is connected to up to $k$ other nodes in the same protein graph that has the smallest Euclidean distance over other nodes, and the distance is smaller than a certain cutoff, i.e., 30  ̊A. These 1-hop neighbors portray the microenvironment of the central node. Furthermore, conducting graph convolutions on this KNN graph aggregates information from the connected neighboring nodes, and the learned hidden embedding represents the local properties of the central node, which we refer to as the micro-frame property of the protein graph geometry.
>
> We have replaced the expression "micro-frame" with "microenvironment" accordingly in the revised version. In the *Introduction*: "Equivariant message passing...encodes AAs’ microenvironment defined by the protein graph’s geometry."; and in the *Conclusion* section: "...message passing layers to extract geometric-aware representation for the microenvironment of central AAs and ...". In Section 2 of defining the protein graph, we specify the concept by "...we first define a symmetric adjacency matrix $A$ with the kNN-graph to capture the nodes' microenvironment...".
>
>
> 3. weakness3 - **invariance and equivariance**:
>
> Thank you for identifying the unprecise expression. We have changed "translation invariance" to "translation equivariance" in the revised version and left "permutation invariance" as it is, according to the definition of SE(3)-equivariance.
>
>
> 4. weakness4 - **dimension of edge features**:
>
> Thank you for reminding the inconsistency about the number of edge features, which should be 93 dimensions with 66 one-hot encoded positions. We have corrected the expression in Appendix A.3 accordingly.
>
> 5. weakness5 - **independence of single mutations**:
>
> We would like to emphasize that our "nonlinear combination" is not reflected in the calculation of the log odd ratio (i.e., the score we used to rank the mutation effects). Instead, we used "a linear summation of individual mutation effects" to describe the fact that most of the existing methods predict AA types one by one, no matter by either masking a particular amino acid or making predictions auto-regressively. The former assumes independence of individual mutations and the latter assumes a conditional probability of a new mutation given all the observed mutations.
>
> Alternatively, for a given (perturbed) protein, LGN predicts the probabilities of each residue's class simultaneously to implement the epistatic effects. While the odd ratios follow the conventional summation rule, these ratios are not independent of each other, i.e., they already embed the joint effect of modifying multiple amino acids. We have modified the expression in contribution 2 to avoid further misunderstanding.
>
>
> 6. weakness6 - **p-values**:
>
> Thank you for the kind suggestion. We have modified the expression accordingly in the revised version to "The $p$-value for both coefficients is $< 0.001$" in the *Experiment* section.
>
>
> 7. weakness7 - **limitation of the work**:
>
> As the potential downside of the research is not compulsory in ICLR submission, we did not include such a discussion in the previous version. We partially agree with the reviewer including MSA information is one critical reason for many baselines to slow down their training and inference speed. However, we argue that **excluding the MSA information is not the only reason that LGN becomes lightweight, and we do not see it as a limitation**. To be specific, LGN outputs the joint distribution of all AA types of the protein, instead of employing a masked model or autoregression. Also, our main algorithm is based on an equivariant graph convolution instead of any variant of the transformer mechanism, which also accelerates the execution speed of the model.

---

### Official Review · Reviewer_cBgt · 2022-10-31

**Confidence:** 5
**Correctness:** 2
**Technical Novelty And Significance:** 3
**Empirical Novelty And Significance:** 2
**Recommendation:** 5

**Clarity, Quality, Novelty And Reproducibility:**

**Clarity**
- “More importantly, when predicting the fitness of the higher-order mutants, most of these models made a crude assumption that the multiple-site mutation effect is a linear summation of the effect of each individual mutation, which is incorrect in most cases” (Section 1) → this is a bit misleading in the context of the prior sentences in that section. Auto-regressive models provide exact likelihood estimation and therefore do not need to resort to these simplifying assumptions with respect to multiple mutants. This is not the case however for architectures relying on masked-language-modeling objectives (eg., ESM-1b / ESM-1v) which do make the individual mutations summation assumption.
- Furthermore, the language in the last paragraph of section 1 would lead one to believe that the method introduced in this paper is not subject to ignoring epistatic effects / do not sum the effects of single mutants independently. However equation 5 in supplementary B.5 seems to indicate the opposite. Could you please clarify?
- The MSA Transformer is not a “zero-shot” but rather a “few shot” method since it does require MSAs at inference to make predictions -- please adjust the language accordingly in section 3.2 & Figure 3
- Figure 3 is a bit difficult to read. Would suggest to separate ablations from comparison with baselines (it is also very hard to tell apart the different ablation points with the current color scheme / symbols). It would also be helpful to add a small table that summarizes average performance comparing your method Vs baselines
- Could you please clarify why a very fast inference time is important in practice? Since the time and cost of in silico experiments is much lower than actual wet labs experiments, should that be a primary concern?

**Quality**
- Benchmark: the set of 15 DMS assays selected in experiments is a subset of the broader ProteinGym benchmark [1] -- why are you not reporting the performance on the full set of assays?
- Baselines: as discussed in the weaknesses section above, to substantiate the claims regarding SOTA made in abstract and conclusion, you should include additional baselines (eg., DeepSequence, EVE, Tranception) and Potts models for the speed of inference claim
- Regarding ESM-1v, are you using a single network or the ensemble of 5 networks as per [2]? The performance of a single non fine-tuned ESM-1v model is relatively low (see Tables 1 and 2 of [2])

**Novelty**
- The multi-task pre-training framework that covers different protein modalities is novel

**Reproducibility**
- Could you please confirm whether the code and data used would be released upon acceptance?

----------------------------------------------------------------------------------------------------------------------------------------

[1] Notin, P., Dias, M., Frazer, J., Marchena-Hurtado, J., Gomez, A.N., Marks, D.S., & Gal, Y. (2022). Tranception: Protein Fitness Prediction with Autoregressive Transformers and Inference-time Retrieval. ICML.

[2] Meier, J., Rao, R., Verkuil, R., Liu, J., Sercu, T., & Rives, A. (2021). Language models enable zero-shot prediction of the effects of mutations on protein function. NeurIPS.


**Strength And Weaknesses:**

**Strengths**
- The GCN layers help bypass data augmentations / be more data efficient
- The multi-task pre-training is a very sensible idea to learn protein embeddings that combine the different modalities characterizing protein sequences (ie., primary, secondary, tertiary structure) and in turn improve downstream task performance (eg., mutation effects prediction)

**Weaknesses**
- As evidenced by the results in section 3.4 and figure 5, there seems to be severe task interference when pre-training simultaneously on all tasks. Including the AA recovery and SAS prediction tasks captures the bulk of the performance lift, while other tasks are either not adding substantial value beyond that, or seem to actually be destructive (eg., dihedral prediction). It is thus not clear whether one should keep all these objectives when pre-training? What is the final training objective / set of pre-training tasks that is recommended to use by the authors?
- The current empirical evaluation lacks several important baselines with respect to mutation effects prediction (eg., DeepSequence, EVE, Tranception) without which claims about “SOTA performance” in abstract/conclusion seem unsubstantiated.
- The method introduced in this work does seem to achieve relatively high fitness prediction performance Vs inference time tradeoff. To drive this point home though would require comparing with Potts models which also achieve a very favorable tradeoff as well.


**Summary Of The Paper:**

This paper focuses on the task of predicting the effects of mutations in protein sequences. To that end it leverages a graph neural network with : 1) Biochemical (eg., AA type, SAS, B-factor) and geometric (eg., 3D coordinates of alpha-carbons, dihedral angles) features of amino acids (nodes in the graphs) and edge features (eg., interatomic distances, local N-C positions and position encoding) 2) A stack of Graph Equivariant Convolution (EGC) layers to learn node and edge embeddings 3) A multi-task pre-training framework that combines AA type classification, SAS and B-factor prediction and 3D-coordinates denoising objectives (positions and angles). The architecture is trained on CATH 4.3.0 and mutation effect prediction is assessed in the zero-shot setting against several baselines.

**Summary Of The Review:**

The modeling approach introduced in this paper is very promising. There are a couple residual issues -- in particular regarding task interference during the multi-task pre-training and evaluation / baselines chosen in experiments -- that would greatly benefit the work if discussed further / corrected. Willing to increase my score if these are addressed during rebuttal.

-------------------------------------------------------------------------------------------------------------------------------------
[Update post rebuttal]
Based on my discussion with the authors and the additional results provided during rebuttal, I believe that the paper in its current form is still not ready for publication and hence maintain my original score (ie., below the acceptance threshold). I provided clear and actionable recommendations to improve the paper in my last response to the authors.

---

> ### Author Response · Authors · 2022-11-14
> **Response to Reviewer cBgt (2)**
>
> 6. clarity3 - **MSA and Few-shot Learning**:
>
> Thank you for pointing out the problem, we have gone through the full text and modified the associated presentation "zero-shot" to "pre-trained" models.
>
> 7. clarity4 - **Figure 3 and a New Table**:
>
> Thank you for the suggestion. We have updated the Figure to display only one of our variants to compare against baseline methods. We also added Table 1 on Page  6 to report the average spearman's correlation of each method on DMS predictions.
>
>
> 8. clarity5 - **Inference Speed**:
>
> We would like to clarify that we do **NOT** claim a very fast inference time as a **primary** concern in algorithm design. While a faster speed is preferred, we agree with the reviewer that the cost for existing *in silico* methods is generally acceptable in comparison to wet lab experiments. In the experiment (Figure 6, for instance) we made the comparison of the inference time for our model against baselines to indirectly advise their gap in training speed. While not all the baseline models have publicly available source code for retraining (for instance, ESM series), and the training resources required are massive, we believe that comparing their inference time is a decent substitute on top of the number of the parameters required and the actual training time reported (in the original papers).
>
>
> 9. quality1 - **ProteinGym Benchmark**:
>
> ProteinGym benchmark covers both single-site and multiple-sites mutations, but our methods are primarily focused on deep mutations. Due to the time limit and for the sake of conciseness, we decided to test more proteins with deep-mutant records and skipped the rest proteins for the comparison models. However, testing on more proteins is a promising route to complete the width of our experiment, and we appreciate the reviewer for pointing this out.
>
>
> 10. quality2 - **baseline methods of inference speed**:
>
> Thank you for the suggestion. We have added the test performance of DeepSequence, Progen2, and Tranception on deep mutations and will add single-mutant results if time permits. The results have been updated to Figure 6 and Table 1-2 with associated explanations in the main text. We regret to skip the Potts models and EVE as explained earlier in Point#3.
>
> 11. quality3 - **different versions of ESM-1v**:
>
> We run all five variants of ESM-1v as provided on their GitHub repository and used the average score over the five. We added the details in Appendix C.2 of the revised version: "...In particular, ESM-1v has 5 variants with different setups and learned parameters, for which we run the test on all the versions and take average performance on them.".
>
>
> 12. reproducibility - **release data and code**:
>
> Thank you for your interest in our implementation. We can confirm that the code and processed protein data will be released publicly upon acceptance.

---

> > ### Comment · Reviewer_cBgt · 2022-12-09
> > **Re: responses from authors**
> >
> > Dear authors,
> >
> > Thank you very much for the thorough responses and additional results.
> >
> > A couple follow up questions / observations based on your updated manuscript:
> >
> > 1. Thank you for the clarification. Just to make sure there is no misunderstanding:
> > a) you obtain the best results on the particular assays chosen in your work (eg. on the 15 assays from Fig3) with AA type, SASA and B-factor. This is fairly robust to the choice of hypers as shown on Fig 5.
> > b) you also illustrate how to integrate the dihedral angle within your framework. Based on these particular 15 assays, integrating dihedral angles leads to a performance decline (as shown on Fig 5), but your point is that it may be useful in other contexts (eg., for different assays). Is that the correct understanding? If yes, could you please provide guidance on when adding dihedral angle is expected to provide a performance lift? Aren't the results in Figure 5 indicative of task interference? (ie. adding a new task to your framework leads to worse overall performance -- this is the main point I was concerned with in my original comment).
> >
> > 2. I greatly appreciate all the extra work done to add new baselines during the rebuttal period. However, the new results in Table 1 seem to be very different from reported results for these baselines on the same assays (see for instance: https://github.com/OATML-Markslab/Tranception/tree/main/proteingym/Detailed_performance_files/Substitutions/Spearman). Could you please comment on the potential discrepancies? These differences are large enough to potentially change the conclusions you drew from these results. Regarding MSAs, they are all publicly available for these assays. Regarding EVE, scoring mutated sequences should be similar to DeepSequence (ie., using the delta ELBO from the corresponding VAE -- no need to use ClinVar labels). EVE has reportedly higher performance than DeepSequence, so it is a useful point of comparison.
> >
> > 3. Fair point regarding MSA compute requirement, but don't you need MSAs to get the AF2 structures used in your model? Do you factor that in the speed assessment or do you assume AF2 structures are a given?
> >
> > 4&5. Thanks for the clarifications. However, the issue you are describing (the independent mutation assumption) only characterizes the models trained with masked-language modeling (eg., ESM-1v, MAS Transformer, ProteinBERT). Neither VAE models (eg., DeepSequence, EVE) nor autoregressive models (eg., Progen, Progen2, RITA, Tranception, ProtGPT2) suffer from that issue -- they both are generative models of the full sequence and do not make the independent mutation assumption. Also note that regarding the point you added earlier in the text ("most of these models made a crude assumption that the mutations on different sites happen sequentially or individually") several autoregressive models do train on / score sequences from N-->C and C-->N (this is the case at least for Progen, Progen2, RITA and Tranception) so the full context from either side of a mutant is taken into account.
> >
> > 6-12. Thank you for the changes and clarifications. No further questions on these points.

---

> > > ### Author Response · Authors · 2022-12-10
> > > **Re: Re: responses from authors**
> > >
> > > We thank the reviewer for making further comments on our response. Regarding the follow-up questions:
> > > 1. Among the five selected prediction tasks (AA, SASA, b-factor, 3D coordinate, and dihedral angle), the former three features are provided by .pdb file from laboratories, and the latter two attributes are from the structure information. In the test set, the 3D structures of the proteins are folded by AlphaFold2, which cannot guarantee 100% accuracy. As a result, we suspect that the performance drop is potentially caused by the structure inaccuracy. When a highly-confident structure is provided, an improved performance would be expected. Otherwise, recovering a less trusty protein structure might mislead the learned embedding.
> > >
> > > 2. (1) We have compared the evaluation step in our program to Tranception's program in https://github.com/OATML-Markslab/Tranception/blob/main/performance_analysis_proteingym.py, and we can confirm that we calculate the Spearman's rho correlation differently. To be specific, Tranception ranks the mutational scores by the different number of mutated AA sites and takes the average on them. The three common proteins (CAPSD_AAV2S, DLG4_HUMAN, and GRB2_HUMAN) have 28, 2, and 2 deepest mutations, respectively. Instead, we calculate an overall spearman correlation no matter the number of mutational sites.
> > > For your reference, in Tranception's code between lines 237 and 272 in `performance_analysis_proteingym.py`, the spearman correlation is calculated by:
> > > >for depth in ['1','2','3','4','5+']:
> > > >&nbsp;&nbsp;&nbsp;&nbsp;...
> > > >&nbsp;&nbsp;&nbsp;&nbsp;performance_DMS['Spearman'] = spearmanr(merged_scores_depth['DMS_score'], merged_scores_depth[score])[0].
> > > >&nbsp;&nbsp;&nbsp;&nbsp;...
> > > >for metric in ['Spearman','AUC','MCC']:
> > > >&nbsp;&nbsp;&nbsp;&nbsp;performance_all_DMS[metric].loc['Average'] = performance_all_DMS[metric].mean().
> > >
> > > As Tranception takes a smaller number of samples to calculate each rank, it is reasonable that their reported scores are higher than ours. We provide a piece of evidence in the Table below, which compares the scores reported by Tranception and us. The difference is calculated by (LGN-Tranception)/Tranception, where positive means LGN's reported score is higher, and negative indicates that Tranception reports a higher correlation score. The negative discrepancies are less common when the depth of mutation is higher (i.e., on CAPSD_AAV2S).
> > > | protein  |   | DeepSequence | Tranception | ProGen2 | MSA-Transformer | ESM-1v |
> > > | -------|--   | ----------- | ----------- | ---------- | ----------- | ---------- |
> > > | CAPSD_AAV2S | LGN | 0.4831 | 0.2307 | 0.2112 | 0.4419 | 0.2212 |
> > > |  |  Tranception  | 0.372  | 0.473  | 0.204  | 0.368  | 0.199  |
> > > |  |  difference     | 0.299  | -0.512 | 0.035  | 0.201  | 0.112  |
> > > | DLG4_HUMAN| LGN  | 0.5013 | 0.6200 | 0.5712 | 0.4654 | 0.4654 |
> > > |    | Tranception  | 0.577  | 0.667  | 0.562  | 0.614  | 0.608  |
> > > |    |difference     | -0.13  | -0.070 | 0.016  | -0.242 | -0.234 |
> > > | GRB2_HUMAN | LGN  | 0.3886 | 0.4441 | 0.5211 | 0.2808 | 0.3211 |
> > > |  |  Tranception  | 0.538  | 0.436  | 0.524  | 0.486  | 0.515  |
> > > |  |  difference     | -0.277 | 0.019  | -0.005 | -0.422 | -0.376 |
> > >
> > >
> > > 2. (2) Regarding the "ClinVar labels" in EVE, we have double-checked the program and their GitHub repository, and we can confirm that the code cannot run on our input protein file. According to the example input file (e.g., https://github.com/OATML/EVE/blob/master/data/labels/ClinVar_labels_P53_PTEN_RASH_SCN5A.csv), ClinVar labels are explicitly required in the input. With this input document, the program runs successfully without any errors. Instead, when feeding one of our protein files to the program, the third step of score prediction prints `ValueError`.
> > >
> > >
> > > 3. We assume the structure information of the test protein is given ahead of running the program for our methods and other structure-based methods (e.g., ESM-IF1 and ProGen2). As such, we did not include the running time for AlphaFold2.
> > >
> > >
> > > 4. &5 We agree with the reviewer that the VAE models can generate the whole protein sequence in one go, and they are potentially free from the independent mutation assumption. We did not make any statement that we are the only work that respects the epistatic effects or the dependency of mutational sites. Instead, we wrote in the paper that "**most** of these models made a crude assumption that...". For the autoregressive models, we would like to clarify again that the dependency that we discussed refers to conditional probability when inferring the AA types. For example, denote P(N1L, H2P) as the probability for a mutant on two-sites N1L, H2P. For autoregressive models, P(N1L, H2P) =PP(H2P｜N1L)P(N1L) (or equivalently log P(N1L, H2P) =log P(H2P｜N1L) + log P(N1L)); instead, LGN (and VAE models) calculate P(N1L, H2P) = P(N1L, H2P) (i.e., log P(N1L, H2P) = log P(N1L, H2P)).
> > >
> > > We hope the explanations above answer the reviewer's confusion.

---

> > > > ### Comment · Reviewer_cBgt · 2022-12-11
> > > > **Re: responses from authors**
> > > >
> > > > Dear authors,
> > > >
> > > > Thank you for the additional responses. While I do believe your paper has several interesting ideas and I enjoyed reading your work, I do believe that the paper in its current form is not yet ready for publication and hence maintain my initial score (ie., below the acceptance threshold).
> > > >
> > > > My main recommendations to the authors in order to improve the paper are as follows:
> > > >
> > > > 1. **Be clearer about which tasks are part of your final model architecture**
> > > >
> > > > Multi-task training can be very finicky -- sometimes additional tasks do not bring value and may even lead to performance decrease (ie., task interference). I would only mention the subset of tasks that are recommended for optimal performance (on test proteins) in the main text, and keep other tasks that were investigated (eg., 3D, dihedral) as ablations in supplement. I found your hypothesis in your last answer re: the dependence on the quality of the structures to be interesting. Perhaps you could validate that hypothesis with additional experiments. I would also encourage you to think through task interference which is another potential reason why you observed the performance drop when adding new tasks (addressing this issue -- if present -- may also further increase the overall performance of your model wrt baselines).
> > > >
> > > > 2. **Correct the performance of reported baselines**
> > > >
> > > > Your last response was incorrect on two accounts:
> > > > a. The reported performance of baselines in ProteinGym is not based on the average spearman by mutation depth as you suggested. It is also computed as the spearman on the overall set of mutations, in the same way that you appear to have done it in your work -- I am absolutely certain about it and checked it myself today (all scores for all baselines for all assays are downloadable from that repo).
> > > > b. About EVE, you only need the delta ELBOs (ie., same as what DeepSequence provides) to compute the desired Spearmans. Following the instructions from the official repo, you get these at the end of step 2 as per the `examples' subfolder -- step 3 is not needed. Here again, I checked it myself today and I am absolutely certain about it.
> > > >
> > > > 3. **Reinforce your analyses and narrative around the key strengths of your model**
> > > >
> > > > As you mention in the abstract, your model appears to achieve a potentially very compelling tradeoff in terms of "spearman on multiple mutants" Vs "speed of inference". I would suggest you to do the following to strengthen that narrative:
> > > > - Add a couple sentences in the introduction to explain why that is important in practice (eg., ability to score all multiple mutants in-silico up to a certain mutation depth in order to guide subsequent experimental validation on the most promising combinations).
> > > > - Add as many multiple mutants assays in your experiments as possible -- right now you have 6, two of which are a bit redundant (F7YBW and F7YBW-MESOW). There are other multiple mutants assays readily available.
> > > > - Add a line item for the aggregated performance on singles (same as last line of Table 1 but for singles, and move assay-specific rows to appendix) -- just to understand whether your method is competitive in that regime (it's really not clear from Fig 3).
> > > > - Split the performance by mutation depth (does your model do better than baselines as mutation depth increases?).
> > > > - Clarify in the caption of Fig 6 the y-axis corresponds to avg Spearman on multiple mutants.
> > > > - Discuss other baselines that also enjoy a very fast inference speed, eg., VESPA/VESPAI from Marquet et al., 2021 (although these models are extremely fast at inference, they only score single variants).
> > > > - As I mentioned in my original review, I believe you should also add simple MSA-based models (eg., site independent, Potts model) as they also have a good "performance Vs inference time" tradeoff. I am very much unconvinced by the argument that extracting an MSA for a protein family is prohibitive from a computational standpoint: a) it is amortized for all mutants computed for that family b) your method is based on AF2 structures for test proteins which itself needs MSAs for prediction (side note: unlike what you stated in your last message, ProGen2 is not structure-based but sequence-only). You could use a sequence-only structure prediction method instead (eg., ESM-fold) but accuracy of predicted structure might be impacted as a result for certain proteins.

---

> ### Author Response · Authors · 2022-11-14
> **Response to Reviewer cBgt (1)**
>
> We thank the reviewer for their detailed comments. Below we answer the addressed concerns point by point.
>
> 1. weakness1 - **Final Training Objective**:
>
> In this paper, we pre-trained our graph on the 31,848 proteins from CATH v4.3.0 that have less than 40% sequence identity and the test proteins are folded with AlphaFold 2. Under this construction, our empirical results show that interpreting the AA type, SASA and B-factor is a solid combination. Meanwhile, we define the objective function (4) to show other available prediction targets, such as the dihedral angle, which might result in better performance with a different training dataset.
>
> 2. weakness2 - **Additional Baselines**:
>
> We thank the reviewer for recommending other baselines. We have implemented DeepSequence (from https://github.com/debbiemarkslab/DeepSequence) and Tranception (from https://github.com/OATML-Markslab/Tranception) on the same task. In addition, we added the third baseline of ProGen2 (from https://github.com/salesforce/progen). Unfortunately, these models usually require a significant amount of time and resources for generating MSA information, and we failed to test the proteins with shallow mutations within such a small period of time. We will supplement those results in the subsequent versions. We updated the results accordingly to our revised version in Figure 3, Figure 6, and Table 1 in the main body.
>
> We could not provide any results by EVE (from https://github.com/OATML/EVE), as there was a key variable called "ClinVar_labels" in the algorithm, which has to be downloaded from http://evemodel.org/. However, we cannot find any of the six proteins that we tested with deep mutants. As we suspect that the proteins were not included in their website, or they have a different naming convention, we will need additional time and effort to investigate the problem and implement the method.
>
> 3. weakness3 - **Comparison to Potts Model**:
>
> We tried to find a valid implementation of the Potts model on GitHub, which however is not implementable in our server. For instance, https://github.com/songlab-cal/factored-attention (and some other repositories) provides a special toolkit that we failed to install on our server; https://github.com/sokrypton/GREMLIN_CPP defines 21 classes of amino acids, while we use 20 for all the models; https://github.com/mstorath/Pottslab does not provide an implementation by Python.  We are trying to make our own implementation for the Potts Model, but at this stage, we cannot make the comparison as advised. We appreciate it if the reviewer could suggest such a toolkit.
>
> Meanwhile, we doubt the speed of inference time for Potts Model, as the efforts in preparing the MSA information should also be included. As defined, Potts Model is an unsupervised method for modeling interactions between amino acids, which leverages the Markov random field to maximize pseudo-likelihood on alignments of proteins that are associated with evolution. When testing the baselines, we find that generating MSA sequences can be extremely painful, which usually takes several hours, not to mention the additional cost of processing the generated sequence.
>
> 4. clarity1 - **Individual Mutations Summation Assumption**:
>
> Thank you for pointing out the misleading presentation in the paper. We used "a linear summation of individual mutation effects" to describe the fact that most of the existing methods predict amino acid types one by one, no matter by either masking a particular amino acid or making predictions auto-regressively. We have changed the words in the second contribution accordingly to avoid further confusion, which now reads: "LGN avoids the independent-mutation assumptions by generating the probabilities of all the amino acid residues at a time, which implements the joint distribution of all variations. In literature, the higher-order mutation effect is usually approached by summing up log-odd-ratio scores of the corresponding individual single-site mutants. The linear combination over separately assigned predictions is unsubstantiated, as the independent mutations neglect the epistatic effect."
>
> 5. clarity2 - **Mutation Effect Acceleration and eq.5**:
>
> As clarified above, our "nonlinear combination" or "respecting the epistatic effects" is not reflected in the calculation of the log odd ratio (that is, the score we used to rank the mutation effects). For a given (perturbed) protein, LGN predicts the probabilities of each residue's class simultaneously to implement the epistatic effects. While the odd ratios follow the conventional summation rule, these ratios are not independent of each other, i.e., they already embed the joint effect of modifying multiple amino acids. In contrast, existing methods generally output residues' change sequentially, which in fact assumes a conditional probability of a new mutation given all the observed mutations.

---

### Decision · Program_Chairs · 2023-01-20

**Decision:**

Reject

**Justification For Why Not Higher Score:**

The reviewers have raised quite a few concerns, including the justification of the technical proposal and the appropriateness of the used metrics, the comprehensiveness of the experimental evaluation, etc. The authors rebuttal failed in convincing the reviewers to change their mind.

**Justification For Why Not Lower Score:**

N/A

**Metareview: Summary, Strengths And Weaknesses:**

This paper focuses on the task of predicting the effects of mutations in protein sequences. A graph neural network is employed, and the model is trained on CATH 4.3.0 and mutation effect prediction is assessed in the zero-shot setting against several baselines.

While this paper works on an interesting topic, the reviewers have raised quite a few concerns, including the justification of the technical proposal and the appropriateness of the used metrics, the comprehensiveness of the experimental evaluation, etc. The authors have provided detailed responses to the review comments, and we have asked the reviewers to reconsider their comments based on the author rebuttal. However, the reviewers end up believing that the author rebuttal did not address their concerns, and would like to keep their original scores. In this case, we do not think it is a good idea to accept the paper.